# Internet addiction and sleep quality among medical students during the COVID-19 pandemic: A multinational cross-sectional survey

Muhammad Junaid Tahir[1,2], Najma Iqbal Malik[3], Irfan Ullah[4], Hamza Rafique Khan[5], Shahida Perveen[3], Rodrigo Ramalho[6], Ahsun Rizwan Siddiqi[7], Summaiya Waheed[8], Mahmoud Mohamed Mohamed Shalaby[9], Domenico De Berardis[10], Samiksha Jain[11], Gautham Lakshmipriya Vetrivendan[12], Harshita Chatterjee[13], William Xochitun Gopar Franco[14], Muhammad Ahsan Shafiq[15], Naira Taiba Fatima[16], Maria Abeysekera[17], Qudsia Sayyeda[18], Shamat Fathi Shamat[19], Wajeeha Aiman[20], Qirat Akhtar[21], Arooj Devi[22], Anam Aftab[8], Sheikh Shoib[23], Chung-Ying Lin[24]*, Amir H. Pakpour[25,26]*

1 Ameer-ud-Din Medical College, Affiliated with University of Health and Sciences, Lahore, Pakistan, 2 Lahore General Hospital, Lahore, Pakistan, 3 University of Sargodha, Sargodha, Pakistan, 4 Kabir Medical College, Gandhara University, Peshawar, Pakistan, 5 Quaid-E-Azam Medical College, Bahawalpur, Pakistan, 6 Department of Social and Community Health, School of Population Health, The University of Auckland, Auckland, New Zealand, 7 Wah Medical College, Affiliated with University of Health Sciences, Wah, Pakistan, 8 Dow Medical College, Dow University of Health Sciences, Karachi, Pakistan, 9 Faculty of Medicine, Ain Shams University, Cairo, Egypt, 10 Department of Mental Health, NHS, Psychiatric Service for Diagnosis and Treatment, Hospital "G. Mazzini", Teramo, Italy, 11 Guntur Medical College, Guntur, Andhra Pradesh, India, 12 Karpaga Vinayaga Institute of Medical Sciences and Research Centre, Maduranthagam, Tamilnadu, India, 13 Texila American University, Georgetown, Guyana, 14 University of Guadalajara, Puerto Vallarta, Mexico, 15 Holy Family Hospital, Rawalpindi, Pakistan, 16 Changsha Medical University, Changsha, Hunan, China, 17 Autonomous University of Santo Domingo, Santo Domingo, Dominica, 18 Red Crescent Clinic Tampa Bay, Tampa, Florida, United States of America, 19 National Ribat University Medical School, Khartoum, Sudan, 20 Beth Israel Deaconess Medical Center, Boston, Massachusetts, United States of America, 21 Aziz Bhatti Shaheed Hospital, Gujrat, Punjab, Pakistan, 22 Ghulam Muhammad Mahar Medical College, Sukkur, Pakistan, 23 Department of Psychiatry, Jawahar Lal Nehru Memorial Hospital, Srinagar, India, 24 Institute of Allied Health Sciences, National Cheng Kung University Hospital, College of Medicine, National Cheng Kung University, Tainan, Taiwan, 25 Department of Nursing, School of Health and Welfare, Jönköping University, Jönköping, Sweden, 26 Social Determinants of Health Research Center, Research Institute for Prevention of Non-Communicable Diseases, Qazvin University of Medical Sciences, ShahidBahounar BLV, Qazvin, Iran

* amir.pakpour@ju.se (AHP); cylin36933@gmail.com (CYL)

**Data Availability Statement:** All the relevant data are within the manuscript and its Supporting information files.

## Abstract

### Background

The emergence of the COVID-19 pandemic has affected the lives of many people, including medical students. The present study explored internet addiction and changes in sleep patterns among medical students during the pandemic and assessed the relationship between them.

### Methods

A cross-sectional study was carried out in seven countries, the Dominican Republic, Egypt, Guyana, India, Mexico, Pakistan, and Sudan, using a convenience sampling technique, an

**Funding:** The authors received no specific funding for this work.

**Competing interests:** The authors have declared that no competing interests exist.

online survey comprising demographic details, information regarding COVID-19, the Pittsburgh Sleep Quality Index (PSQI), and the Internet Addiction Test (IAT).

## Results

In total, 2749 participants completed the questionnaire. Of the total, 67.6% scored above 30 in the IAT, suggesting the presence of an Internet addiction, and 73.5% scored equal and above 5 in the PSQI, suggesting poor sleep quality. Internet addiction was found to be significant predictors of poor sleep quality, causing 13.2% of the variance in poor sleep quality. Participants who reported COVID-19 related symptoms had disturbed sleep and higher internet addiction levels when compared with those who did not. Participants who reported a diagnosis of COVID-19 reported poor sleep quality. Those living with a COVID-19 diagnosed patient reported higher internet addiction and worse sleep quality compared with those who did not have any COVID-19 patients in their surroundings.

## Conclusion

The results of this study suggest that internet addiction and poor sleep quality are two issues that require addressing amongst medical students. Medical training institutions should do their best to minimize their negative impact, particularly during the current COVID-19 pandemic.

## Background

The internet has completely revolutionized the world in the past few decades, with the 21st century witnessing explosive growth in worldwide internet usage [1]. This global digitalization has provided better opportunities for education, communication, banking, businesses, health-seeking, and social interaction [1]. Unfortunately, uncontrolled use of internet may lead to maladaptive behaviors [2, 3]. One of the maladaptive behaviors is internet addiction (or termed as problematic internet use or pathological internet use [2] with the definition of "excessive or poorly controlled preoccupations, urges or behaviors regarding computer use and internet access that leads to impairment or distress" [4]. Indeed, evidence shows that internet addiction was linked to other psychiatric disorders (e.g., attention deficit and hyperactivity and alcohol abuse) [5]. Thus, prolonged Internet usage may adversely impact both physical and mental health.

Sleep problems had a clear correlation with mental health, psychiatric illnesses, and disorders related to anxiety and mood [6–9]. Moreover, Cellini et al. [10] found that spending ample time on internet was associated with poor sleep quality and may further lead to an increase of psychological distress (i.e., stress, anxiety and depression) among young adults. Additionally, internet addiction resulted in the dysfunction of daily activities, including neglect of household chores and reduced productivity [10]. Therefore, one can tentatively conclude that pathological internet usage can also negatively affect the circadian rhythm causing insomnia and other sleep disturbances [11–14].

However, the increasing popularity of smartphones makes the use of a smartphone before sleep has become a habit for students [15]. Moreover, it was found that teenagers who had trouble falling asleep or sustaining sleep were progressively inclined to have an internet addiction, and people who were dependent on the internet had their basic circadian rhythms altered

[16]. Clear-cut impact that lockdown pose on sleep disorders is understudied, although evidence shows that the students spent more time on digital devices before sleep and had irregular sleep pattern, which may lead to poor sleep quality [17]. Therefore, sleep and internet addiction are of concern: Insufficient sleep and poor sleep quality may result in poor memory and weakened learning abilities, which jeopardize the academic performance of students and can also result in other growth and developmental disorders [18–20]. Excessive internet usage may lead to grey matter atrophy, which negatively affects one's ability to concentrate and hinders their decision-making capacity [21].

The COVID-19 pandemic has led to an inevitable surge in the use of digital technologies due to the physical distancing norms and nationwide lockdowns, including medical schools [22, 23]. Medical schools in both developed and developing countries have utilized modern technologies to bring effective changes in medical education [24]. Over the years, particularly during COVID-19 pandemic, medical students have faced significant changes in their education (e.g., online classes and virtual learning) [25–27]. Due to the decrease in face-to-face social interactions and the increased time spent indoors during COVID-19 pandemic, there is a growing dependence on social media and online entertainment platforms for social interaction [28, 29]. This increased online learning may contribute to internet addiction. Apart from the issue of online learning, prior evidence shows that age [30] gender [6, 17, 30], family system [17], year of MBBS [17], smoking history [30], and physical health status [6, 30] were associated with internet addiction.

Empirically it was evident that excessive internet usage may significantly affect the sleep cycle of the person which leads to insomnia, irregular sleep patterns and excessive daytime sleepiness [14, 31]. Current study evaluated the effect of COVID-19 pandemic on the use of internet and subsequently, sleeping habits among the medical students. It also viewed at how the sleeping habits were altered by the excessive or problematic use of internet. To date, relatively few studies have examined the effect of the current COVID-19 pandemic on internet addiction and sleep quality particularly among medical students. University graduates from all around the world reported a common problem. Azad et al. [6] compared these sleep related issues among medical students and economics and law undergraduates. He mentioned that medical students rank the highest on the prevalence regarding poor sleep and even worse quality of life in comparison of students from other subject groups. Matter of fact that segregate these medical graduates from other peer groups included their overall lifestyle, attitude towards study and more academic pressure. Therefore, it is important to understand internet addiction and sleep among medical students.

This study was thus aimed to investigate the problem of internet addiction and sleep globally; primary focus was to select countries from all continents. However, availability of researcher and collaborators ended up with selection of these countries finally from North America (Mexico, Dominican Republic), South America (Guyana), Africa (Egypt, Sudan) and Asia (Pakistan, India). In these continents, further these countries were identified on basis of statistics provided by WHO regarding COVID-19. The selected countries were on higher risk due to poor health facilities, extreme poverty and low SGD's indicator. Moreover, COVID-19 statistics of WHO updated on march 2020 revealed higher percentage of COVID-19 confirmed and suspected cases in these regions. The present study aims to fill this gap. With this regard, several hypotheses were made below: (i) internet addiction is positively associated with poor sleep quality among medical students in different countries; (ii) subdomains in the internet addiction are significant predictors for poor sleep quality among medical students in different countries; (iii) students with issues related to COVID-19 have high levels of internet addiction and poor sleep quality among medical students in different countries; and (iv)

students with different demographics have different levels of internet addiction and sleep quality among medical students in different countries.

## Methods

### Study participants

A survey research design was applied, using a non-probability convenience sampling technique. The research team developed a consent form and a questionnaire using shared Google® forms. Data collection took place between June and July 2020 after study approval from the University of Sargodha, Pakistan. The target population was students aged 15 to 44 years old in medical colleges from the Dominican Republic, Egypt, Guyana, India, Mexico, Pakistan, and Sudan. First, the relevant institutions' concerned authorities provided their informed consent, authorizing the study to be conducted with students who attended their institutions. Then, potential participants were contacted directly by the researcher through their active email addresses and WhatsApp® numbers provided by their institute co-coordinators. A web link to the questionnaire was then shared with them, and informed consent was obtained before participating in the study. The bulk of the data collection was from countries which delivered their curriculum in English and they could understand English language easily. Hence, responses from these places were collected in an English version of questionnaire. Translation to Spanish was done for collecting data from only two countries i.e., Dominican Republic and Mexico, in order to reduce the language barrier in their countries. For these two countries, a single questionnaire (file attached) containing both languages were circulated, and participants could read the question in English or Spanish according to their ease. Participants were not given any monetary benefits for participating in the study. To restrict duplication of responses, respondents were asked to provide email address, as only one response can be submitted through one email address. After data collection, prior to data analysis, data was reviewed and cleaned incomplete responses were discarded and only complete responses per subject were analyzed further for hypotheses testing. The ethic committee of University of Sargodha approved the study procedure (SU/PSY/786-S, April 09,2020). Consent was informed by all participants. All participants provided online informed consent. Regarding sample size calculation, guidelines for being more statistically sound about sample size tells sample of 10% of population size is recommend until sample size become 1000. A general rule of thumb is that larger sample size will increase the generalizability of the results; therefore, we tried to keep sample size as large as possible.

### Measure

The questionnaire bank included demographic questions and two scales, the Pittsburgh Sleep Quality Index (PSQI) and the Internet Addiction Test (IAT) [32, 33]. The consent form and the questionnaire were developed in two languages, English and Spanish.

**Demographics information.**   Collected demographics information included age, gender, country of residence, whether they lived in an urban or rural area, family system (joint or nuclear family), whether the participant attended a private or a public medical university, and the participants' year of study and smoking history; plus, the following COVID-19 related questions: (i) do you have any COVID-19 related symptoms? (ii) have you been diagnosed with COVID-19 by a health professional? (iii) have you been living with a person diagnosed with COVID-19? (iv) are you following COVID-19 standard operating procedures (SOPs)?

**Pittsburgh Sleep Quality Index (PSQI).**   PSQI is a self-rated questionnaire [32] that assesses sleep quality and disturbances over a 1-month time interval through 19 individual items, four open-ended questions, and 14 questions rated on a Likert scale from 0–3 with 3

reflecting the extreme negative. These 19 items generate seven component scores, i.e., subjective sleep quality, sleep latency, sleep duration, habitual sleep efficiency, sleep disturbances, use of sleeping medication, and daytime dysfunction, and the sum of scores for these seven components yields a PSQI global score of sleep quality which ranges from 0 to 21 [34]. A total sum of 5 or more indicates "poor" quality of sleep [34]. The scale has acceptable reliability, Cronbach's α of 0.914 [35] and 0.73 [36]. The Reliability of the PSQI in present was excellent (Cronbach's α of 0.91). Raniti et al. [36] validated the single factor structure of PSQI in an adolescent sample and mentioned covariation among poor sleep efficiency, latency and poor duration within this age students which is considerable.

**Internet Addiction Test (IAT).**    The IAT is a 20-item questionnaire that measures characteristics and behaviors associated with compulsive use of the internet [33]. Each item is weighted along a Likert-scale continuum that ranges from 0 = less extreme behavior to 5 = most extreme behavior. Total scores that range from 0 to 30 points are read as representing a normal level of internet usage, a score range of 31 to 49 indicates the presence of a mild level of internet addiction, scores of 50 to 79 indicate a moderate level, and scores of 80 to 100 indicate a severe level of internet addiction. IAT has high Cronbach's α of 0.93 [37]. The reliability coefficient of the IAT was excellent in present study (Cronbach's α = 0.77). Significant correlations of subscales confirm the internal consistency of the scales used in this study. Further, this scale consisted of 5 subscales, whose reliability was also calculated. They were salience (Cronbach's α = 0.77), excessive use (Cronbach's α = 0.77), neglect work (Cronbach's α = 0.77), anticipation (Cronbach's α = 0.77), lack of control (Cronbach's α = 0.77), neglect social life. (Cronbach's α = 0.77).

**Statistical analysis.**    All collected data were scored and analyzed. Descriptive statistics including mean, standard deviation (SD), range, skewness, alpha reliability coefficient of all scales and their subscales were computed. Further, mean, standard deviation, range, and skewness were computed for the PSQI and IAT, including their total score and subscale scores. The mean differences in PSQI and IAT (including total and subscale scores) between demographic variables were explored and compared using independent t-tests and effect size (i.e., Cohen's *d*). Multiple linear regression and multiple logistic models were constructed to examine the associations between sleep and internet addiction; and the effects of predictors on sleep and internet addiction. More specifically, the total scores of the PSQI and IAT were used to define having a sleep problem and having internet addiction; then, multiple logistic regressions were used for the PSQI and IAT total scores with their cutoffs. The domain scores of the PSQI and IAT do not have a cutoff score, and multiple linear regressions were used for the domain scores. For all the linear and logistic regression models, demographic variables (including age [adolescent vs. adult], gender [male vs. female] residence [rural vs. urban], family system [joint vs. nuclear], medical university sector [public vs. private], year of MBBS [junior vs. senior], smoking history [yes vs. no], health status [poor vs. good], COVID-19 related symptoms [yes vs. no], COVID-19 diagnosis [yes vs. no], live with COVID-19 infected individuals [yes vs. no], and COVID-19 standard operating procedures [yes vs. no]) were included.

## Results

The final sample size was comprised of 2749 participants, 991 (36%) male and 1758 (64%) female participants. Further demographic information is presented in Table 1. A majority of the participants were from Pakistan (n = 1009; 36.7%) and India (n = 939; 34.2%). Most participants, 2311 (84.1%), resided in urban areas, and 1936 (70.4%) belonged to a nuclear family system. Also, most participants studied at a public university, 1678 (61%), and 2035 (74%) were juniors (1st year to 4th year) and 714 (26%) seniors (5th and 6th year). Regarding the COVID-

**Table 1. Participants' characteristics with comparisons of IAT and PSQI (N = 2749).**

| | n (%) | IAT | | | PSQI | | |
|---|---|---|---|---|---|---|---|
| | | Mean (SD) | 95% CI | Cohen's *d* | Mean (SD) | 95% CI | Cohen's *d* |
| **Age** | | | | | | | |
| Adolescent (15–20) | 1065 (38.7) | 40.46 (17.78) | -0.18, -0.04 | 0.07 | 6.47 (3.14) | -0.18, -0.67 | 0.13 |
| Adult (21–44) | 1684 (61.3) | 39.15 (17.86) | | | 6.89 (3.18) | | |
| **Gender** | | | | | | | |
| Male | 991 (36.0) | 40.98 (17.48) | 3.47, 0.69 | 0.11 | 6.61 (3.15) | 0.06, 0.69 | 0.05 |
| Female | 1758 (64.0) | 38.90 (17.99) | | | 6.79 (3.17) | | |
| **Residence** | | | | | | | |
| Rural | 438 (15.9) | 38.10 (18.79) | -0.01, -3.65 | 0.10 | 6.21 (3.45) | -0.26, -0.96 | 0.18 |
| Urban | 2311 (84.1) | 39.94 (17.64) | | | 6.82 (3.10) | | |
| **Family system** | | | | | | | |
| Joint family | 812 (29.6) | 41.19 (18.62) | 3.65, 0.73 | 0.12 | 6.99 (3.36) | 0.64, 0.11 | 0.11 |
| Nuclear family | 1936 (70.4) | 39.00 (17.46) | | | 6.62 (3.07) | | |
| **University sector** | | | | | | | |
| Public | 1678 (61.0) | 38.19 (17.85) | -2.38, -5.11 | 0.21 | 6.52 (3.13) | -0.29, -0.77 | 0.17 |
| Private | 1071 (39.0) | 41.94 (17.58) | | | 7.06 (3.19) | | |
| **Year of MBBS** | | | | | | | |
| Junior | 2035 (74.0) | 40.66 (17.83) | 5.40, 2.37 | 0.21 | 6.65 (3.19) | -0.01, -0.55 | 0.09 |
| Senior | 714 (26.0) | 36.77 (17.56) | | | 6.94 (3.07) | | |
| **Smoking history** | | | | | | | |
| Yes | 272 (9.9) | 42.62 (16.86) | 5.52, 1.06 | 0.18 | 7.61 (3.37) | 1.37, 0.58 | 0.30 |
| No | 2477 (90.1) | 39.32 (17.91) | | | 6.63 (3.13) | | |
| **Health status** | | | | | | | |
| Poor | 1048 (38.1) | 44.24 (17.08) | 8.76, 6.07 | 0.42 | 7.63 (3.17) | 1.69, 1.21 | 0.47 |
| Good | 1701 (61.9) | 36.82 (17.71) | | | 6.17 (3.03) | | |
| **COVID-19 symptoms** | | | | | | | |
| Yes | 125 (4.5) | 42.94 (17.34) | 6.64, 0.24 | 0.19 | 8.08 (3.33) | 1.97, 0.84 | 0.43 |
| No | 2621 (95.5) | 39.49 (17.85) | | | 6.66 (3.14) | | |
| **COVID-19 diagnosis** | | | | | | | |
| Yes | 189 (6.9) | 41.77 (15.74) | 6.33, -1.98 | 0.12 | 7.94 (3.63) | 2.06, 0.42 | 0.36 |
| No | 2690 (93.1) | 39.60 (17.88) | | | 6.70 (3.15) | | |
| **Live with COVID-19 infected individual** | | | | | | | |
| Yes | 189 (6.9) | 43.59 (16.14) | 6.64, 1.81 | 0.24 | 8.04 (3.16) | 1.87, 0.94 | 0.44 |
| No | 2560 (93.1) | 39.36 (17.92) | | | 6.63 (3.14) | | |
| **Follow COVID-19 instructions** | | | | | | | |
| Yes | 2398 (87.2) | 39.46 (17.69) | 0.53, -3.45 | 0.08 | 6.70 (3.12) | 0.16, -0.60 | 0.06 |
| No | 351 (12.8) | 40.92 (18.78) | | | 6.92 (3.48) | | |

Note: IAT = Internet Addiction Test; PSQI = Pittsburgh Sleep Quality Index.

19 related questions, most participants, 2398 (87.2%) reported following COVID-19 SOPs. Similarly, most participants, 2621 (95.5%), reported no COVID-19 related symptoms, and only 189 (6.9%) had been diagnosed with COVID-19 by a health professional.

Overall, 67.6% (n = 1859) of the sample scored above 30 in the IAT, indicating the presence of an internet addiction. Among the 1859 participants; 40.0% (n = 1099) scored between 31 and 49 in the IAT, indicating the presence of a mild level internet addiction; 25.5% (n = 700) scored between 50 and 79, indicating a moderate level internet addiction; and 2.2% (n = 60)

**Table 2. Descriptive information for the Pittsburgh Sleep Quality Index (PSQI), Internet Addiction Test (IAT), and their subscales (N = 2749).**

| Scales or subscales | Mean | SD | Potential range | Actual range | Skewness |
|---|---|---|---|---|---|
| **PSQI global** | 6.73 | 3.16 | 0–21 | 0–19 | 0.43 |
| Subjective sleep quality | 1.09 | 0.81 | 0–3 | 0–3 | 0.46 |
| Sleep latency | 0.80 | 0.88 | 0–3 | 0–3 | 0.17 |
| Sleep duration | 0.80 | 0.88 | 0–3 | 0–3 | 0.17 |
| Sleep efficiency | 1.05 | 1.12 | 0–3 | 0–3 | 0.90 |
| Sleep disturbance | 1.14 | 0.54 | 0–3 | 0–3 | 0.82 |
| Use of sleep medication | 0.19 | 0.55 | 0–3 | 0–3 | 0.29 |
| Daytime dysfunction | 1.07 | 0.82 | 0–3 | 0–3 | 0.36 |
| **IAT total** | 39.65 | 17.84 | 0–100 | 0–100 | 0.31 |
| Salience | 9.10 | 5.16 | 0–25 | 0–25 | 0.42 |
| Excessive use | 10.82 | 4.96 | 0–25 | 0–25 | 0.25 |
| Neglect work | 5.84 | 3.64 | 0–15 | 0–15 | 0.38 |
| Anticipation | 3.68 | 2.32 | 0–10 | 0–10 | 0.48 |
| Lack of control | 7.16 | 3.63 | 0–15 | 0–15 | 0.15 |
| Neglect social life | 3.02 | 2.08 | 0–10 | 0–10 | 0.64 |

scored 80 and over, indicating a severe dependence. Additionally, 73.5% (n = 2007) of the total sample had a global score of 5 or more on the PSQI, indicating poor sleep quality. Detailed scores in PSQI and IAT are presented in Table 2.

The correlations between PSQI and IAT were found to be significant (r = 0.36; p <0.01). Moreover, Table 3 reports that significant predictors for internet addiction included university sector (AOR = 1.35; 95% CI = 1.13, 1.62), year of MBBS (AOR = 0.74, 95% CI = 0.59, 0.94), smoking history (AOR = 0.73, 95% CI = 0.54, 0.97), and health status (AOR = 0.59, 95% CI = 0.49, 0.70). Table 4 reports that significant predictors for sleep included residence (AOR = 1.50, 95% CI = 1.16, 1.81), health status (AOR = 0.56, 95% CI = 0.47, 0.67), COVID-19 related symptoms (AOR = 0.62, 95% CI = 0.39, 0.99), living with COVID-19 infected individuals (AOR = 0.58, 95% CI = 0.40, 0.85), salience in internet addiction (AOR = 1.03, 95% CI = 1.00, 1.06), and excessive use in internet addiction (AOR = 1.12, 95% CI = 1.09, 1.15).

## Discussion

The present study explored some aspects of the lifestyle (i.e., internet addiction and sleep) among medical college students across seven different countries during the period of COVID-19 pandemic. The prevalence rates of internet addiction and poor sleep were relatively high in the medical students during COVID-19 pandemic. Moreover, the present study found that university sector, year of MBBS, smoking history, and health status were significant predictors for internet addiction. Residence, health status, COVID-19 related symptoms, living with COVID-19 infected individuals, salience in internet addiction, and excessive use in internet addiction were significant predictors for poor sleep.

As previously reported by Garcia-Priego [38], public health emergencies can cause, trigger, or worsen mental health concerns, plus, they can also be related to a high prevalence of low to mild levels of Internet addiction. Regarding COVID-19, a study by Siste et. al. [39] found that mental health issues and sleep disruptions were related to Internet addiction, and it was especially prevalent in groups with proximity to COVID-19. Fear of COVID-19 contraction and rampant misinformation about COVID-19 could have contributed to these results.

**Table 3. Predictors on internet addiction (N = 2749).**

| IV | Dependent variable of internet addiction: AOR (95%CI)[a] or B (SE)[b] | | | | | | |
|---|---|---|---|---|---|---|---|
| | IAT total[a] | Sal.[b] | EU[b] | NW[b] | Anti.[b] | LoC[b] | NSL[b] |
| Age | 0.96(0.78,1.18) | 0.29(0.22) | 0.13(0.22) | -0.40(0.16) | -0.07(0.10) | -0.27(0.16) | -0.011(0.09)* |
| Gender | 0.94(0.78,1.14) | -0.58(0.21)** | -0.20(0.20) | -0.24(0.15) | 0.004(0.10) | 0.14(0.15) | -0.63(0.08) |
| Residence | 1.08(0.84,1.38) | 0.24(0.27) | 0.77(0.26)** | 0.34(0.19) | -0.06(0.12) | 0.33(0.19) | 0.02(0.11)* |
| Family | 0.89(0.73,1.08) | -0.57(0.22)** | -0.40(0.21)* | 0.03(0.15) | -0.25(0.10)* | -0.30(0.15) | -0.11(0.09) |
| University | 1.35(1.13,1.62) | 0.86(0.20)** | 0.66(0.19)* | 0.36(0.14) | 0.32(0.09)** | 0.24(0.14) | 0.36(0.08) |
| MBBS | 0.74(0.59,0.94) | -0.92(0.25)** | -0.29(0.24) | -0.86(0.18)** | -0.29(0.12)* | -0.72(0.18)** | -0.15(0.10) |
| Smoking | 0.73(0.54,0.97) | -0.42(0.33) | -10.12(0.32)* | 0.09(0.23) | -0.20(0.15) | -0.02(0.24) | -0.54(0.13) |
| Health status | 0.59(0.49,0.70) | -1.56(0.20)** | -1.76(0.19)** | -1.43(0.22)** | -0.56(0.09)** | -1.10(0.14) | -0.40(0.08) |
| Symptom | 1.00(0.64,1.54) | -0.31(0.48) | -0.61(0.46) | -0.08(0.34) | -0.44(0.22)* | -0.30(0.34) | 0.05(0.19)* |
| Diagnosis | 1.71(0.84,3.48) | 0.03(0.71) | 1.27(0.68) | -0.24(0.50) | -0.29(0.32) | -0.29(0.50) | 0.17(0.28)* |
| Live | 0.76(0.54,1.09) | -1.63(0.40)** | -0.75(0.38)* | -0.31(0.22) | -0.13(0.18) | -0.23(0.28) | -0.75(0.16) |
| Instruction | 1.15(0.89,1.50) | 0.21(0.29) | 0.10(0.28) | 0.33(0.20) | 0.13(0.13) | -0.16(0.21) | -0.01(0.12)* |

Note. IV = independent variable; PSQI = Pittsburgh Sleep Quality Index; SSQ = subjective sleep quality; SL = sleep latency; S.DUR = sleep duration; SE = sleep efficiency; S.DIST = sleep disturbance; UoSM = use of sleep medicine; D.DYS = daytime dysfunction; Sal. = salience; EU = excessive use; NW = neglect work; Anti. = anticipation; LoC = lack of control; NSL = neglect social life; Family = family system; University = university sector; MBBS = year of MBBS; Smoking = smoking history; Symptom = COVID-19-related symptom; Diagnosis = COVID-19 diagnosis; Live = live with COVID-19 infected individuals; Instruction = follow COVID-19 instructions.

Reference groups were: age (adolescents); gender (males); residence (rural); family system (joint); university sector (public); year of MBBS (junior); smoking history (yes); health status (poor); COVID-19 related symptoms (no); COVID-19 diagnosis (no); live with COVID-19 infected individuals (no); and follow COVID-19 instructions (no).

[a] Reported in adjusted odds ratio (AOR) with 95% confidence interval (CI) by logistic regression model.

[b] Reported in unstandardized coefficient with standard error (SE) by linear regression model.

In the present study, adolescents and male participants scored higher on Internet addiction when compared to adults and female participants respectively. However, both male and female were non significantly different in terms of their sleep quality. Similar results were reported by Kim et al. [40]. Similar results were found in other studies too [6, 17, 30], where individuals with internet addiction tended to be males of young age and found no significant gender difference in sleep quality. The authors further reported that adults with internet addiction reported high difficulty in initiating and maintaining sleep, had a non-restorative sleep cycle, showed daytime functional impairment, and their duration of sleep was less than 10 hours on weekdays.

The present study found no significant effect of the place of residence on internet addiction. However, it did have a significant effect on sleep quality, as participants living in urban areas had poorer sleep quality when compared to those living in rural areas. Participants living in joint families and those attending a private medical university scored higher on internet addiction and poorer sleep quality when compared to those living in nuclear families and attending a public medical university, respectively. Some opposite results were reported by Jahan et al. [41], who found that living with their family was associated with better sleep quality. In the present study, juniors scored higher on Internet addiction than seniors, while senior students reported poorer sleep quality. Here, a similar result was previously reported by Cheng et al. [42], and Romero-Blanco et al. [17], who found that undergraduates (although not separated in junior or senior years) had poorer sleep quality than postgraduates.

Participants with a smoking history and self-reported poor health scored higher on Internet addiction and also reported poor sleep quality when compared to those who did not smoke

**Table 4. Predictors on sleep quality (N = 2749).**

| IV | Dependent variable of sleep quality: AOR (95%CI)[a] or B (SE)[b] | | | | | | | |
|---|---|---|---|---|---|---|---|---|
| | PSQI global[a] | SSQ[b] | SL[b] | SE[b] | S.DUR[b] | S.DIST[b] | UoSM[b] | D.DYS[b] |
| Age | 1.19(0.98,1.44) | 0.04(0.03) | 0.001(0.04) | -0.02(0.05) | 0.09(0.04)* | 0.05(0.02)* | 0.06(0.02)* | -0.003(0.03) |
| Gender | 1.18(0.98,1.41) | 0.01(0.03) | 0.11(0.04) | 0.17(0.05)** | -0.02(0.04) | 0.05(0.02)* | 0.001(0.02) | 0.06(0.03)* |
| Residence | 1.50(1.16,1.81) | 0.09(0.04) | 0.13(0.05) | 0.16(0.06)* | -0.04(0.05) | 0.003(0.03) | -0.04(0.03) | 0.03(0.04) |
| Family | 1.08(0.90,1.30) | -0.04(0.03) | 0.07(0.04) | 0.03(0.05) | -0.06(0.04) | -0.03(0.02) | -0.04(0.02) | -0.04(0.03) |
| University | 1.21(0.94,1.30) | 0.07(0.03)* | 0.01(0.04) | -0.04(0.05) | 0.05(0.04) | 0.04(0.02) | 0.01(0.02) | 0.09(0.03) |
| MBBS | 1.15(0.92,1.43) | 0.07(0.04) | 0.09(0.05) | 0.10(0.06) | 0.05(0.04) | -0.003 (0.03) | 0.02(0.03) | -0.07(0.04) |
| Smoking | 0.77(0.57,1.03) | -0.09(0.05) | -0.03(0.06) | -0.22 (0.07)* | -0.008(0.06) | -0.10(0.03)** | -0.10(0.04)** | -0.08(0.05) |
| Health status | 0.56(0.47,0.67) | -0.24(0.03)** | -0.24(0.04)** | -0.07(0.05) | -0.05(0.04) | -0.11 (0.02)** | -0.05(0.02)* | -0.20(0.03)** |
| Symptom | 0.62(0.39,0.99) | -0.13(0.07) | 0.08(0.09) | -0.14(0.11) | 0.01(0.08) | -0.19 (0.05)** | -0.16(0.05)** | -0.15(0.07)* |
| Diagnosis | 0.66(0.33,1.31) | -0.14(0.11) | -0.12(0.13) | 0.03(0.16) | -0.08(0.12) | -0.11(0.07) | -0.15(0.08) | 0.05(0.11) |
| Live | 0.58(0.40,0.85) | -0.10(0.06) | -0.06(0.08) | 0.09(0.09) | -0.17(0.07) | 0.16(0.04)** | -0.13(0.04)** | -0.24(0.06) |
| Instruction | 1.12(0.87,1.44) | 0.02(0.04) | 0.02(0.06) | -0.02(0.07) | 0.07(0.05) | 0.01(0.03) | 0.13(0.03)** | 0.01(0.04) |
| Sal. | 1.03(1.00,1.06) | -0.004(0.005) | 0.01(0.006) | -0.002(0.007) | -0.003(0.005) | 0.01(0.003)** | 0.01(0.003)** | 0.01(0.005) |
| EU | 1.12(1.09,1.15) | 0.04(0.005)** | 0.05(0.006)** | 0.001(0.008) | 0.03(0.006)** | 0.002(0.003) | 0.005(0.004) | 0.03(0.005)** |
| NW | 0.99(0.96,1.02) | 0.002(0.006) | -0.02(0.007)* | -0.007(0.009) | -0.01(0.01) | 0.009(0.004)* | 0.002(0.004) | 0.04(0.006)** |
| Anti. | 1.01(0.96,1.06) | 0.005(0.008) | 0.005(0.01) | -0.01(0.01) | 0.01(0.01) | 0.02(0.06)** | 0.005(0.006) | 0.01(0.01) |
| LoC | 0.99(0.96,1.03) | -0.004(0.006) | 0.0004(0.008) | 0.02(0.01) | -0.01(0.01) | -0.003(0.004) | -0.02(0.004)** | 0.01(0.01) |
| NSL | 1.02(0.97,1.07) | 0.02(0.008) | 0.001(0.01) | 0.01(0.001) | 0.01(0.01) | 0.02(0.006)** | 0.03(0.006)** | 0.002(0.008) |

Note. IV = independent variable; PSQI = Pittsburgh Sleep Quality Index; SSQ = subjective sleep quality; SL = sleep latency; S.DUR = sleep duration; SE = sleep efficiency; S.DIST = sleep disturbance; UoSM = use of sleep medicine; D.DYS = daytime dysfunction; Sal. = salience; EU = excessive use; NW = neglect work; Anti. = anticipation; LoC = lack of control; NSL = neglect social life; Family = family system; University = university sector; MBBS = year of MBBS; Smoking = smoking history; Symptom = COVID-19-related symptom; Diagnosis = COVID-19 diagnosis; Live = live with COVID-19 infected individuals; Instruction = follow COVID-19 instructions.

Reference groups were: age (adolescents); gender (males); residence (rural); family system (joint); university sector (public); year of MBBS (junior); smoking history (yes); health status (poor); COVID-19 related symptoms (no); COVID-19 diagnosis (no); live with COVID-19 infected individuals (no); and follow COVID-19 instructions (no).

[a] Reported in adjusted odds ratio (AOR) with 95% confidence interval (CI) by logistic regression model.

[b] Reported in unstandardized coefficient with standard error (SE) by linear regression model.

and reported good health. Previous studies found similar results in this regard. For example, Jahan et al. [41] also found that non-smokers had a better sleep quality. A study conducted in Taiwan found that students who smoked had altered sleeping patterns and were less likely to have good sleep quality. Liu et al. [42] and Cheng et al. [43] also reported that people who reported poor sleep quality also smoked excessive and indulged in high internet surfing.

Results suggest that the IAT and all its subscales have a significant positive correlation with PSQI, and all its subscales, except sleep efficiency, with which it had a non-significant relationship. In summary, internet addiction was a significant predictor of poor sleep quality. Previous studies have also found similar results. Jahan et al. [41] study with medical college students of Bangladesh also found as internet addiction increased, the level of poor sleep quality also increased. The study by Canan et. al. [14], conducted with Turkish high school students, also reported an association between internet addiction and impaired sleep. Similarly, the study by Tsitsika [44], conducted in seven European countries, found that the prevalence of sleep problems was higher among students with an internet addiction. Finally, a systematic review conducted by Lam [12], found that internet gaming addiction and problematic internet use were

related to sleep problems, including insomnia and poor sleep quality, although more so the latter than the former.

The coronavirus has caused an extraordinary crisis in many fields; therefore, many countries are struggling to get out of the crisis. Mulyadi et al. [45] examine the relationship between sleep duration and anxiety and internet usage duration. Their results showed a significant relationship between sleep duration and anxiety; and internet usage duration with anxiety. Another study from the Middle East region (Kuwait, Saudi Arabia) reported by Alheneidi et al. [46] showed an association between loneliness and problematic internet use (PIU), and an association between loneliness and the number of hours spent online. Those who reported greater loneliness also obtained frequent news about the pandemic from social media. Their ANOVA analyses further showed a dose-response between the predictors and PIU. Moreover, Lin [47] mentioned that the COVID-19 outbreak has significantly disrupted normal activities globally. During this epidemic, people around the world were expected to encounter several mental health challenges. Internet addiction may become a serious issue among teenagers. The prevalence of internet addiction was found to be 24.4% during this period [47]. Additionally, previous studies report an increase in gaming addiction and internet use with detrimental impact on psychosocial well-being. Fernandes et al. [48] compared the impact of lockdown on internet use among adolescents via their habits between before and during the pandemic. They found the relationship between gaming addiction, internet use and COVID-19 worries and showed that adolescents generally have increased their use of social media sites and streaming services. Further, those who had a high level of gaming addiction, compulsive internet use and social media use also reported high scores of depression, loneliness, escapism, poor sleep quality and anxiety related to the pandemic. Their findings indicated that, regardless of country of residence, the COVID-19 outbreak has had a significant effect on adolescent internet use and psychosocial well-being.

The present study also found the varied effect of different constructs of internet addiction on different constructs of sleep quality. For example, anticipation, a subscale of the IAT, was a significant predictor of sleep disturbance, and lack of control was a significant predictor of the use of sleep medication. Neglecting social life, a domain of the IAT, was a significant predictor of sleep disturbance, use of sleep medication, and subjective sleep quality. Work neglect, a sub-construct of IAT, was a significant predictor of sleep disturbance, sleep duration, and daytime dysfunction. Lastly, the salience of internet addiction and excessive use of the internet (subscales of IAT) were significant predictors of sleep disturbance, use of sleep medication, subjective sleep quality, sleep duration, daytime dysfunction, and sleep latency. A previous study conducted by Lin et al. [30] also reported that internet addiction was significantly associated with subjective sleep disturbance, use of sleep medication, sleep quality, sleep duration, daytime dysfunction, and sleep latency.

The strength of this study is that it has focused on the area of research where not much work has been done during the COVID-19 pandemic. Other strengths include the large sample size, use of an international sample (increasing generalizability), and the potentially important implications for mental health and sleep interventions among medical students. However, our study had several limitations as follows: Firstly, the convenience sampling method was used and due to voluntary participation, there was a possibility of selection bias. Specifically, those who did not have internet addiction or sleep problems were more likely than those who had such a problem to agree to participate. Therefore, the results of the present study may underestimate the prevalence of internet addiction and sleep problems. Secondly, because this was a cross sectional study, we were unable to establish causal inferences. That is, it is unclear whether internet addiction results in sleep problems, or the other way around. Future studies are thus needed to use a longitudinal design or case-control design to provide causal

relationship evidence. Thirdly, most participants belonged to Pakistan and India due to which it is difficult to generalize the results. Specifically, the present study's results may be prone more to Pakistani and Indians instead of other countries' participants. Fourthly, the varying prevalence of COVID-19 in different countries accompanied with the subsequent imposition of SOPs and the degree of lockdown individually, could have been a contributor to bias and was not taken into account. Finally, the data collection was via the online mode; therefore, those who did not frequently surf on their internet during the COVID-19 pandemic might not be aware of this study. This may restrict the generalizability of the present findings. That is, those who were at very low level of internet addiction or internet use might not participate in the present study.

Looking at the results, future studies should focus on how to allow the medical students to use internet in a balanced manner so as to prevent the development of internet addiction during such lockdown conditions and, devise measures to improve the quality of sleep among medical students.

## Conclusion

The present study partially supports the hypotheses that during COVID-19 pandemic period, internet addiction was positively associated with poor sleep among medical students; salience and excessive use in the internet addiction were significant predictors for poor sleep among medical students; medical students had high levels of internet addiction and poor sleep during COVID-19 pandemic; and some demographic characteristics were associated with internet addiction and poor sleep among medical students. More specifically, the study found that the prevalence of internet addiction was about 67.6% and that of poor sleep was 73.5%. The presence of COVID-19 related symptoms was associated with disturbed sleep and higher scores in the IAT, and a diagnosis of COVID-19 was associated with poor sleep quality. Similarly, living with someone with a COVID-19 diagnosis was associated with a higher score on the IAT and worse sleep quality. These findings suggest the importance of providing medical students with coping strategies that would prevent pathological Internet usage and poor sleep quality. This study highlights the need to design some sort of training to deal with such pandemic situation, which was previously not given to student sample.

## Supporting information

**S1 File. Dataset.**
(SAV)

## Author Contributions

**Conceptualization:** Muhammad Junaid Tahir, Irfan Ullah, Hamza Rafique Khan, Mahmoud Mohamed Mohamed Shalaby.

**Data curation:** Muhammad Junaid Tahir, Najma Iqbal Malik, Irfan Ullah, Mahmoud Mohamed Mohamed Shalaby, Sheikh Shoib.

**Formal analysis:** Muhammad Junaid Tahir, Najma Iqbal Malik, Irfan Ullah, Shahida Perveen, Rodrigo Ramalho, Ahsun Rizwan Siddiqi, Sheikh Shoib, Amir H. Pakpour.

**Investigation:** Najma Iqbal Malik, Irfan Ullah, Hamza Rafique Khan, Shahida Perveen, Rodrigo Ramalho, Ahsun Rizwan Siddiqi, Summaiya Waheed, Domenico De Berardis, Samiksha Jain, Gautham Lakshmipriya Vetrivendan, Harshita Chatterjee, William Xochitun Gopar Franco, Muhammad Ahsan Shafiq, Naira Taiba Fatima, Maria Abeysekera,

Qudsia Sayyeda, Shamat Fathi Shamat, Wajeeha Aiman, Qirat Akhtar, Arooj Devi, Sheikh Shoib, Chung-Ying Lin.

**Methodology:** Irfan Ullah, Hamza Rafique Khan, Shahida Perveen, Ahsun Rizwan Siddiqi, Summaiya Waheed, Domenico De Berardis, Samiksha Jain, Gautham Lakshmipriya Vetrivendan, Harshita Chatterjee, William Xochitun Gopar Franco, Muhammad Ahsan Shafiq, Naira Taiba Fatima, Maria Abeysekera, Qudsia Sayyeda, Shamat Fathi Shamat, Wajeeha Aiman, Qirat Akhtar, Arooj Devi, Sheikh Shoib, Chung-Ying Lin, Amir H. Pakpour.

**Project administration:** Irfan Ullah, Samiksha Jain, Wajeeha Aiman, Qirat Akhtar, Arooj Devi, Anam Aftab, Sheikh Shoib.

**Resources:** Hamza Rafique Khan, Rodrigo Ramalho, Summaiya Waheed, Mahmoud Mohamed Mohamed Shalaby, Chung-Ying Lin.

**Software:** Shahida Perveen, Summaiya Waheed, Anam Aftab.

**Supervision:** Chung-Ying Lin, Amir H. Pakpour.

**Validation:** Ahsun Rizwan Siddiqi, Qudsia Sayyeda, Anam Aftab, Amir H. Pakpour.

**Visualization:** Qudsia Sayyeda.

**Writing – review & editing:** Amir H. Pakpour.

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
