## [Decision Letter · Decision Letter 0]

16 Aug 2021

PONE-D-21-19421

Internet addiction and sleep quality among medical students during the COVID-19 pandemic: A multinational cross-sectional survey

PLOS ONE

Dear Dr. Pakpour,

Thank you for submitting your manuscript to PLOS ONE. After careful consideration, we feel that it has merit but does not fully meet PLOS ONE’s publication criteria as it currently stands. Therefore, we invite you to submit a revised version of the manuscript that addresses the points raised during the review process.

We look forward to receiving your revised manuscript.

Kind regards,

Forough Mortazavi

Academic Editor

PLOS ONE

Journal Requirements:

Additional Editor Comments (if provided):

Dear Authors,

According to the study title, it is expected that Internet addiction and sleep quality and predictors of both variables be investigated among students; however, the authors rely on simple tests such as t-test rather than complicated tests which could enable them to disclose predictors of the two variables.The statistical analysis presented by the authors is mostly not relevant to the objectives of study. This makes the study uninteresting and reduces its usefulness.  The English language of the manuscript is not up to standard. I hope my comments would help the authors improve their manuscript.

Background:

1. Although this section is too long, no proper explanation is provided for conducting this study in several countries and the reasons for selecting the specific countries. PLS summarize this section and add the necessary explanations.

2. In page 10, the English in these sentences requires correction: “Palatty et al. [27] further differentiated economics and Law graduates had more stressful academic demands and responsibilities in their study programs in comparison to students belong to various other academic discipline. This fact explained the reason of poor sleep among medical students compared to other student population.”

3. The declared objectives of the study are too general. PLS define them in such a way as to justify data collection from several countries.

Methods

4. According to the authors, the global score for sleep quality is 0-21. This seems wrong because the score is calculated based on 14 items rated on a Likert scale from 0-3. PLS provide a reference for the cutoff point of 5 for poor quality of sleep.

5. PLS describe scoring of the 5 open-ended items and the calculation of the total score for the PSQI.

6. The details of sample size calculation are not presented.

7. You have conducted multiple regression analyses assessing the associations between sleep and Internet addiction. Please explain why you did not conduct multiple logistic regression because as you mentioned in page 13, the scale has a cut-off value.

Results

8. This section needs extensive revision.

9. Table 1 is redundant because its contents are already presented in table 3 except for the percentages which can be included in table 3.

10. The relationships between family size and internet addiction and sleep quality are valuable; however, what is important and should be emphasized is that the family size is a predictor of internet addiction and sleep quality. In studies such as this the focus must be on the results of multiple regression analysis. Table 4 is also redundant because it presents the results of t-test and not the regression.

11. In table 5, no comparisons between countries are presented. It seems that the author' aim was to collect a large amount of data without comparing the countries. There are no discussions of the results presented in table 5 therefore; we regard this tale as redundant.

12. The range of correlations between two constructs can be presented in the text. PLS remove table 6, its contents bear no relation to the aims of the study.

13. In table 6. The authors controlled for the effects of other variables to determine if internet addiction impacted sleep quality; however in discussing the results presented in the table they cover several other risk factors of sleep quality and internet addiction. In my opinion, considering the large amount of data, the authors must determine predictors of both poor sleep quality and internet addiction. If these variables have cut-off values, PLS conduct multiple logistic regressions.

14. I suggest that the results of table 6 be presented fully; so that, readers can estimate the effects of all significant variables on poor sleep quality.

15. With such a large data set, it is expected and valuable to determine associated factors and predictors of poor sleep quality and internet addiction.

Discussion

16. This manuscript is difficult to follow. It contains so many redundant tables and so much information that the authors cannot manage to discuss fully. In the discussion section, they mostly focus on the results of table 3.

17. PLS provide a summary of significant results in the first paragraph.

18. In page 17, you write, “To compare cutoff with previous cutoff scores, percentile ranks were calculated and results identified no major differences in score range.” This sentence is vague. What do you mean by “previous cutoff scores and percentile ranks.”

19. PLS provide the reference for this sentence: “Superficially, minimum value of range was slightly shifted from 40 to 51 in the total score of IAT. However, the cutoff score is within the range of 40-69.

20. The last lines of the first paragraph and the second paragraph are more suitable for the results section. PLS do not repeat the results in the discussion section.

21. In page 18, paragraph 2, the duration of sleep less than 10 hours should be reported, “their duration of sleep was more than 10 hours on weekdays.”

22. Page 19, “Additionally, this study also indicated significant mean differences on all scales and subscales other than sleep latency, sleep efficiency, and daytime dysfunction.” It is not clear what is meant by this sentence.

23. PLS correct wrong usages of capital letter in the text.

Conclusions

24. This section includes conclusions unsupported by results: In the first lines, the authors write that, “The present study suggests the possibility that the COVID-19 pandemic, along with its public health measures, has had a significant impact on Internet use and sleep quality amongst medical students.” The objective of this study is not a comparison of pre pandemic and pandemic Internet use and sleep quality. The above statement cannot be presented as a conclusion of you study.

25. Furthermore, the authors mentioned that, “psychological mindedness was impacted due to COVID-19 because student did not expected and trained to handle this situation.” This is not investigated by this study too.

26. The last sentence cannot also be concluded from the study.

Reviewers' comments:

Reviewer's Responses to Questions

**Comments to the Author**

1. Is the manuscript technically sound, and do the data support the conclusions?

Reviewer #1: Yes

Reviewer #2: Partly

2. Has the statistical analysis been performed appropriately and rigorously? 

Reviewer #1: Yes

Reviewer #2: No

3. Have the authors made all data underlying the findings in their manuscript fully available?

Reviewer #1: No

Reviewer #2: Yes

4. Is the manuscript presented in an intelligible fashion and written in standard English?

Reviewer #1: Yes

Reviewer #2: Yes

5. Review Comments to the Author

Reviewer #1: Firstl,this is a very good sleep data covering a number of countries.Internet addiction was investigated among medical students and sample size is large enough.There are several changes that need to be made.

1.In the discussion part, there is no data support for the results discussed that need to add.

2.There are too many tables in this paper, so I need to provide some statistical graphs.

3.The presentation in Table 7 is not important and needs to be revised.

Reviewer #2: I understood that this study was an epidemiological study that analyzed the relationship between Internet dependence and various parameters of sleep using the method of Internet survey among medical students in several countries.

One of the features of this study was that it was interesting to examine the effects of the presence of COVID-19 patients on Internet dependence and sleep.

However, I think that this study has some serious problems that cannot be ignored in order to be published in an international journal.

The biggest problem is that there are serious questions about the representativeness and reliability of the sample used for the survey. One of the major drawbacks of Internet-based surveys is that the population bias of the population from which the data is collected is assumed to be quite large. Since the survey was conducted using the Internet, there is a high possibility that the data will be biased toward the population that originally spent a lot of effort on the Internet. Also, in this study, there is no information on how many people were invited to participate in the survey. The response rate of the survey is important information for estimating the magnitude of bias in the reporting of such epidemiological studies (if the response rate is low, the nonresponse bias is quite low, and the reliability of the survey is low).

For these reasons, we believe that the value of this study can be enhanced by publishing it in a domestic journal rather than in an international journal.

6. PLOS authors have the option to publish the peer review history of their article (what does this mean?). If published, this will include your full peer review and any attached files.

Reviewer #1: No

Reviewer #2: No

---

## [Author Response · Author response to Decision Letter 0]

15 Oct 2021

August 18, 2021

Dear Dr. Mortazavi,

Thanks for giving us the opportunity to revise our work “Internet addiction and sleep quality among medical students during the COVID-19 pandemic: A multinational cross-sectional survey (Manuscript ID PONE-D-21-19421)”. After revising the manuscript, we have resubmitted it to be considered for publication on the PLOS ONE. We have systematically addressed the reviewers’ concerns point-by-point, and the revisions are presented in red font in the manuscript. We hope that the paper is now acceptable for publication in the PLOS ONE.

We look forward to your reply. Thank you for considering our manuscript.

Sincerely,

Corresponding Author

Responses to Editor:

Background:

1. Although this section is too long, no proper explanation is provided for conducting this study in several countries and the reasons for selecting the specific countries. PLS summarize this section and add the necessary explanations.

Response: We have now made the Background section concise (from 1250 words to 923 words) and provided the reasons why the study was conducted in these countries. Specifically, the aim of conducting an international study was to study the impact of the pandemic on medical students globally. The reason for selecting these countries in particular was a rather approachable means of data collection and recruitment of collaborators from these countries. We collected data using convenience and snowball methods. 

“This study was thus aimed to investigate the problem of internet addiction and sleep globally; primary focus was to select countries from all continents. However, availability of researcher and collaborators ended up with selection of these countries finally from North America (Mexico, Dominican Republic), South America (Guyana), Africa (Egypt, Sudan) and Asia (Pakistan, India). In these continents, further these countries were identified on basis of statistics provided by WHO regarding COVID-19. The selected countries were on higher risk due to poor health facilities, extreme poverty and low SGD’s indicator. Moreover, COVID-19 statistics of WHO updated on march 2020 revealed higher percentage of COVID-19 confirmed and suspected cases in these regions.”

2. In page 10, the English in these sentences requires correction: “Palatty et al. [27] further differentiated economics and Law graduates had more stressful academic demands and responsibilities in their study programs in comparison to students belong to various other academic discipline. This fact explained the reason of poor sleep among medical students compared to other student population.”

Response: We have now deleted these sentences as they are not relevant to the present study’s population.

3. The declared objectives of the study are too general. PLS define them in such a way as to justify data collection from several countries.

Response: We have now revised the study objectives.

“The present study aims to fill this gap. With this regard, several hypotheses were made below: (i) internet addiction is positively associated with poor sleep quality among medical students in different countries; (ii) subdomains in the internet addiction are significant predictors for poor sleep quality among medical students in different countries; (iii) students with issues related to COVID-19 have high levels of internet addiction and poor sleep quality among medical students in different countries; and (iv) students with different demographics have different levels of internet addiction and sleep quality among medical students in different countries.”

Methods

4. According to the authors, the global score for sleep quality is 0-21. This seems wrong because the score is calculated based on 14 items rated on a Likert scale from 0-3. PLS provide a reference for the cutoff point of 5 for poor quality of sleep.

Response: The scoring of Pittsburgh Sleep Quality Index (PSQI) is quite complicated. Therefore, we have now provided the information here (in the response letter) instead of putting the complicated information in the manuscript to avoid distraction to the potential readers. However, we have cited the reference in the manuscript for readers to understand the scoring method.

The PSQI 19 items were reformed into seven component scores (each component score ranges between 0 and 3). The 0-21 global score is the sum of the seven component scores. 

Component 1 score = PSQI item 9 Score

Component 2 score = PSQI item 2 Score (<15min (0), 16-30min (1), 31-60 min (2), >60min (3)) + PSQI item 5a Score (if sum is equal 0=0; 1-2=1; 3-4=2; 5-6=3) 

Component 3 score = PSQI item 4 Score (>7(0), 6-7 (1), 5-6 (2), <5 (3) 

Component 4 score = (total # of hours asleep) / (total # of hours in bed) x 100; then, recoded into >85%=0, 75%-84%=1, 65%-74%=2, <65%=3 (Note: the hours asleep information is derived from PSQI item 4; the hours in bed information is derived from PSQI items 1 and 3)

Component 5 score = sum of scores of PSQI items 5b to 5j; then, recoded into 0=0; 1-9=1; 10-18=2; 19-27=3.

Component 6 score = PSQI item 6 Score 

Component 7 score = sum of PSIQ item 7 Score + PSQI item 8 score; then, recoded into 0=0; 1-2=1; 3-4=2; 5-6=3 

Detailed PSQI items and scoring can be found at here: http://www.goodmedicine.org.uk/files/assessment,%20pittsburgh%20psqi.pdf

Reference (for both cutoff at 5 and scoring method): 

Buysse,D.J., Reynolds,C.F., Monk,T.H., Berman,S.R., & Kupfer,D.J. (1989). The Pittsburgh Sleep Quality Index (PSQI): A new instrument for psychiatric research and practice. Psychiatry Research, 28(2), 193-213.

5. PLS describe scoring of the 5 open-ended items and the calculation of the total score for the PSQI.

Response: Please see our response to the previous comment. 

6. The details of sample size calculation are not presented.

Response: We have now provided the information how we calculated the sample size. 

“Regarding sample size calculation, guidelines for being more statistically sound about sample size tells sample of 10% of population size is recommend until sample size become 1000. A general rule of thumb is that larger sample size will increase the generalizability of the results; therefore, we tried to keep sample size as large as possible.”

7. You have conducted multiple regression analyses assessing the associations between sleep and Internet addiction. Please explain why you did not conduct multiple logistic regression because as you mentioned in page 13, the scale has a cut-off value.

Response: We have now used multiple logistic regression when appropriate for the data analysis. The tables have been rearranged and updated. 

“Multiple linear regression and multiple logistic models were constructed to examine the associations between sleep and internet addiction; and the effects of predictors on sleep and internet addiction. More specifically, the total scores of the PSQI and IAT were used to define having a sleep problem and having internet addiction; then, multiple logistic regressions were used for the PSQI and IAT total scores with their cutoffs. The domain scores of the PSQI and IAT do not have a cutoff score, and multiple linear regressions were used for the domain scores.”

Results

8. This section needs extensive revision.

Response: We have now revised the section substantially according to your comments.

“The final sample size was comprised of 2749 participants, 991 (36%) male and 1758 (64%) female participants. Further demographic information is presented in Table 1. A majority of the participants were from Pakistan (n=1009; 36.7%) and India (n=939; 34.2%). Most participants, 2311(84.1%), resided in urban areas, and 1936 (70.4%) belonged to a nuclear family system. Also, most participants studied at a public university, 1678 (61%), and 2035 (74%) were juniors (1st year to 4th year) and 714 (26%) seniors (5th and 6th year). Regarding the COVID-19 related questions, most participants, 2398 (87.2%) reported following COVID-19 SOPs. Similarly, most participants, 2621 (95.5%), reported no COVID-19 related symptoms, and only 189 (6.9%) had been diagnosed with COVID-19 by a health professional.

Overall, 67.6% (n = 1859) of the sample scored above 30 in the IAT, indicating the presence of an Internet addiction. Among the 1859 participants; 40.0% (n = 1099) scored between 31 and 49 in the IAT, indicating the presence of a mild level Internet addiction; 25.5% (n = 700) scored between 50 and 79, indicating a moderate level Internet addiction; and 2.2% (n= 60) scored 80 and over, indicating a severe dependence. Additionally, 73.5% (n = 2007) of the total sample had a global score of 5 or more on the PSQI, indicating poor sleep quality. Detailed scores in PSQI and IAT are presented in Table 2.

The correlations between PSQI and IAT were found to be significant (r = 0.36; p <0.01). Moreover, Table 3 reports that significant predictors for internet addiction included university sector (AOR = 1.35; 95% CI = 1.13, 1.62), year of MBBS (AOR = 0.74, 95% CI = 0.59, 0.94), smoking history (AOR = 0.73, 95% CI = 0.54, 0.97), and health status (AOR = 0.59, 95% CI = 0.49, 0.70). Table 4 reports that significant predictors for sleep included residence (AOR = 1.50, 95% CI = 1.16, 1.81), health status (AOR = 0.56, 95% CI = 0.47, 0.67), COVID-19 related symptoms (AOR = 0.62, 95% CI = 0.39, 0.99), living with COVID-19 infected individuals (AOR = 0.58, 95% CI = 0.40, 0.85), salience in internet addiction (AOR = 1.03, 95% CI = 1.00, 1.06), and excessive use in internet addiction (AOR = 1.12, 95% CI = 1.09, 1.15).”

9. Table 1 is redundant because its contents are already presented in table 3 except for the percentages which can be included in table 3.

Response: We have now combined Tables 1 and 3. 

10. The relationships between family size and internet addiction and sleep quality are valuable; however, what is important and should be emphasized is that the family size is a predictor of internet addiction and sleep quality. In studies such as this the focus must be on the results of multiple regression analysis. Table 4 is also redundant because it presents the results of t-test and not the regression.

Response: We have now removed Table 4 and provided a table on regression analysis results. 

11. In table 5, no comparisons between countries are presented. It seems that the author' aim was to collect a large amount of data without comparing the countries. There are no discussions of the results presented in table 5 therefore; we regard this tale as redundant.

Response: We have now removed Table 5. 

12. The range of correlations between two constructs can be presented in the text. PLS remove table 6, its contents bear no relation to the aims of the study.

Response: We have now removed Table 6 and reported the correlation between PSQI and IAT scores in the Results section.

13. In table 6. The authors controlled for the effects of other variables to determine if internet addiction impacted sleep quality; however in discussing the results presented in the table they cover several other risk factors of sleep quality and internet addiction. In my opinion, considering the large amount of data, the authors must determine predictors of both poor sleep quality and internet addiction. If these variables have cut-off values, PLS conduct multiple logistic regressions.

Response: We have now revised our statistical methods and rewritten the Results section. 

14. I suggest that the results of table 7 be presented fully; so that, readers can estimate the effects of all significant variables on poor sleep quality.

Response: We have now reported all the independent variables in the regression models.

15. With such a large data set, it is expected and valuable to determine associated factors and predictors of poor sleep quality and internet addiction.

Response: We have now conducted multiple logistic regression models to assess potential predictors for poor sleep quality and internet addiction, separately. 

Discussion

16. This manuscript is difficult to follow. It contains so many redundant tables and so much information that the authors cannot manage to discuss fully. In the discussion section, they mostly focus on the results of table 3.

Response: Thank you for the constructive comments. We have now revised the Discussion based on the revised Results. 

17. PLS provide a summary of significant results in the first paragraph.

Response: We have now provided a summary of significant findings in the first paragraph of the Discussion. 

“The present study explored the impact of COVID-19 on the lifestyle of medical college students across seven different countries. The prevalence rates of internet addiction and poor sleep were relatively high in the medical students during COVID-19 pandemic. Moreover, the present study found that university sector, year of MBBS, smoking history, and health status were significant predictors for internet addiction. Residence, health status, COVID-19 related symptoms, living with COVID-19 infected individuals, salience in internet addiction, and excessive use in internet addiction were significant predictors for poor sleep.”

18. In page 17, you write, “To compare cutoff with previous cutoff scores, percentile ranks were calculated and results identified no major differences in score range.” This sentence is vague. What do you mean by “previous cutoff scores and percentile ranks.”

Response: We have now deleted this sentence. 

19. PLS provide the reference for this sentence: “Superficially, minimum value of range was slightly shifted from 40 to 51 in the total score of IAT. However, the cutoff score is within the range of 40-69.

Response: Thank you. This is a typo; the cutoff should be 50-79 instead of 40-69. 

In our sample the sample score on 75th percentile was calculated as 51. Which means that internet addiction is prevalent at the starting limit of moderate addiction (scores range from 50-79 as described by Young (1998, 2011).

20. The last lines of the first paragraph and the second paragraph are more suitable for the results section. PLS do not repeat the results in the discussion section.

Response: We have now deleted the two paragraphs. 

21. In page 18, paragraph 2, the duration of sleep less than 10 hours should be reported, “their duration of sleep was more than 10 hours on weekdays.”

Response: We have now changed “more than 10 hours” to “less than 10 hours”. Thank you for catching this typo. 

22. Page 19, “Additionally, this study also indicated significant mean differences on all scales and subscales other than sleep latency, sleep efficiency, and daytime dysfunction.” It is not clear what is meant by this sentence.

Response: We have now deleted the sentence. 

23. PLS correct wrong usages of capital letter in the text.

Response: The incorrect uses of capital letters were revised. 

Conclusions

24. This section includes conclusions unsupported by results: In the first lines, the authors write that, “The present study suggests the possibility that the COVID-19 pandemic, along with its public health measures, has had a significant impact on Internet use and sleep quality amongst medical students.” The objective of this study is not a comparison of pre pandemic and pandemic Internet use and sleep quality. The above statement cannot be presented as a conclusion of you study.

Response: We have now revised our conclusion. 

“The present study partially supports the hypotheses that during COVID-19 pandemic period, internet addiction was positively associated with poor sleep among medical students; salience and excessive use in the internet addiction were significant predictors for poor sleep among medical students; medical students had high levels of internet addiction and poor sleep during COVID-19 pandemic; and some demographic characteristics were associated with internet addiction and poor sleep among medical students. More specifically, the study found that the prevalence of internet addiction was about 67.6% and that of poor sleep was 73.5%. The presence of COVID-19 related symptoms was associated with disturbed sleep and higher scores in the IAT, and a diagnosis of COVID-19 was associated with poor sleep quality. Similarly, living with someone with a COVID-19 diagnosis was associated with a higher score on the IAT and worse sleep quality. These findings suggest the importance of providing medical students with coping strategies that would prevent pathological Internet usage and poor sleep quality. This study highlights the need to design some sort of training to deal with such pandemic situation, which was previously not given to student sample.”

25. Furthermore, the authors mentioned that, “psychological mindedness was impacted due to COVID-19 because student did not expected and trained to handle this situation.” This is not investigated by this study too.

Response: The sentence is deleted. 

26. The last sentence cannot also be concluded from the study. 

Response: The last sentence of the Conclusion is deleted. 

Responses to Reviewer #1:

First of all, this is a very good sleep data covering a number of countries. Internet addiction was investigated among medical students and sample size is large enough. There are several changes that need to be made.

1. In the discussion part, there is no data support for the results discussed that need to add.

Response: Thank you for the positive comment on our manuscript. We also appreciate the specific comments made which helped us to strengthen our paper. 

2. There are too many tables in this paper, so I need to provide some statistical graphs.

Response: We have now reduced the number of tables in the revised manuscript. 

3. The presentation in Table 7 is not important and needs to be revised. 

Response: We have now revised Table 7. 

Responses to Reviewer #2:

1. I understood that this study was an epidemiological study that analyzed the relationship between Internet dependence and various parameters of sleep using the method of Internet survey among medical students in several countries.

One of the features of this study was that it was interesting to examine the effects of the presence of COVID-19 patients on Internet dependence and sleep.

However, I think that this study has some serious problems that cannot be ignored in order to be published in an international journal.

Response: Thank you for the positive comment on our manuscript. We also appreciate the specific comments made which helped us to strengthen our paper. 

2. The biggest problem is that there are serious questions about the representativeness and reliability of the sample used for the survey. One of the major drawbacks of Internet-based surveys is that the population bias of the population from which the data is collected is assumed to be quite large. Since the survey was conducted using the Internet, there is a high possibility that the data will be biased toward the population that originally spent a lot of effort on the Internet. Also, in this study, there is no information on how many people were invited to participate in the survey. The response rate of the survey is important information for estimating the magnitude of bias in the reporting of such epidemiological studies (if the response rate is low, the nonresponse bias is quite low, and the reliability of the survey is low).

Response: As the anonymized survey was used so we were unable track response rate 100% however, at start of questionnaire, a question given “are you willing to participate in this study? “54 answered “No”, while 2749 answered “yes” and filled complete questionnaire. 

3. For these reasons, we believe that the value of this study can be enhanced by publishing it in a domestic journal rather than in an international journal. 

Response: We have now substantially revised the manuscript based on the constructive comments made by the editor and reviewers. Therefore, we believe that our study is valuable in an international journal.

---

## [Decision Letter · Decision Letter 1]

18 Oct 2021

PONE-D-21-19421R1Internet addiction and sleep quality among medical students during the COVID-19 pandemic: A multinational cross-sectional surveyPLOS ONE

Dear Dr. Pakpour,

Thank you for submitting your manuscript to PLOS ONE. After careful consideration, we feel that it has merit but does not fully meet PLOS ONE’s publication criteria as it currently stands. Therefore, we invite you to submit a revised version of the manuscript that addresses the points raised during the review process.

We look forward to receiving your revised manuscript.

Kind regards,

Forough Mortazavi

Academic Editor

PLOS ONE

Journal Requirements:

Additional Editor Comments:

Dear authors,

Thank you for revising the manuscript and submitting it to PLOS ONE. A few things remain to be fixed but it has improved by the changes already made. We invite you to submit a revised version of the manuscript that addresses the points below:

The first sentence of the discussion section: “The present study explored the impact of COVID-19 on the lifestyle of medical college students across seven different countries.”

I do not agree with this statement. PLS kindly explain how this study explored the impact of the covid-19 on students’ lifestyle.

Please kindly discuss limitations of the study and potential sources of bias or imprecision. Discuss both direction and magnitude of any potential bias.

Please discuss the generalizability of the study results.

Good Luck

Reviewers' comments:

Reviewer's Responses to Questions

**Comments to the Author**

1. If the authors have adequately addressed your comments raised in a previous round of review and you feel that this manuscript is now acceptable for publication, you may indicate that here to bypass the “Comments to the Author” section, enter your conflict of interest statement in the “Confidential to Editor” section, and submit your "Accept" recommendation.

Reviewer #1: All comments have been addressed

2. Is the manuscript technically sound, and do the data support the conclusions?

Reviewer #1: Yes

3. Has the statistical analysis been performed appropriately and rigorously? 

Reviewer #1: Yes

4. Have the authors made all data underlying the findings in their manuscript fully available?

Reviewer #1: No

5. Is the manuscript presented in an intelligible fashion and written in standard English?

Reviewer #1: No

6. Review Comments to the Author

Reviewer #1: The language of this manuscript needs further improvement.The quality control of network questionnaire has not been further elaborated.Please list the method in this manuscript.

7. PLOS authors have the option to publish the peer review history of their article (what does this mean?). If published, this will include your full peer review and any attached files.

Reviewer #1: No

---

## [Author Response · Author response to Decision Letter 1]

19 Oct 2021

October 19, 2021

Dear Dr. Mortazavi,

Thanks for giving us the opportunity to revise our work “Internet addiction and sleep quality among medical students during the COVID-19 pandemic: A multinational cross-sectional survey (Manuscript ID PONE-D-21-19421R1)”. After revising the manuscript, we have resubmitted it to be considered for publication on the PLOS ONE. We have systematically addressed the reviewers’ concerns point-by-point, and the revisions are presented in red font in the manuscript. We hope that the paper is now acceptable for publication in the PLOS ONE.

We look forward to your reply. Thank you for considering our manuscript.

Sincerely,

Corresponding Authors

Responses to Editor:

Background:

1. Thank you for revising the manuscript and submitting it to PLOS ONE. A few things remain to be fixed but it has improved by the changes already made. We invite you to submit a revised version of the manuscript that addresses the points below:

Response: Thank you for appreciating our previous revisions and thank you again for providing us additional constructive comments below to improve our work.

2. The first sentence of the discussion section: “The present study explored the impact of COVID-19 on the lifestyle of medical college students across seven different countries.”

I do not agree with this statement. PLS kindly explain how this study explored the impact of the covid-19 on students’ lifestyle.

Please kindly discuss limitations of the study and potential sources of bias or imprecision. Discuss both direction and magnitude of any potential bias.

Please discuss the generalizability of the study results.

Response: We have now revised the first sentence of the Discussion to precisely reflect what the present study investigates. That is, we have removed the claim of “impact” in the sentence as we did not have pre- and post-pandemic information to understand “impact”.

“The present study explored some aspects of the lifestyle (i.e., internet addiction and sleep) among medical college students across seven different countries during the period of COVID-19 pandemic.”

3. Please kindly discuss limitations of the study and potential sources of bias or imprecision. Discuss both direction and magnitude of any potential bias.

Please discuss the generalizability of the study results.

Response: Limitations of the study, including the potential sources of bias/imprecision, and the generalizability of the study results have been added.

“However, our study had several limitations as follows: Firstly, the convenience sampling method was used and due to voluntary participation, there was a possibility of selection bias. Specifically, those who did not have internet addiction or sleep problems were more likely than those who had such a problem to agree to participate. Therefore, the results of the present study may underestimate the prevalence of internet addiction and sleep problems. Secondly, because this was a cross sectional study, we were unable to establish causal inferences. That is, it is unclear whether internet addiction results in sleep problems, or the other way around. Future studies are thus needed to use a longitudinal design or case-control design to provide causal relationship evidence. Thirdly, most participants belonged to Pakistan and India due to which it is difficult to generalize the results. Specifically, the present study’s results may be prone more to Pakistani and Indians instead of other countries’ participants. Fourthly, the varying prevalence of COVID-19 in different countries accompanied with the subsequent imposition of SOPs and the degree of lockdown individually, could have been a contributor to bias and was not taken into account. Finally, the data collection was via the online mode; therefore, those who did not frequently surf on their internet during the COVID-19 pandemic might not be aware of this study. This may restrict the generalizability of the present findings. That is, those who were at very low level of internet addiction or internet use might not participate in the present study.”

---

## [Editor Report · Decision Letter 2]

22 Oct 2021

Internet addiction and sleep quality among medical students during the COVID-19 pandemic: A multinational cross-sectional survey

PONE-D-21-19421R2

Dear Dr. Pakpour,

We’re pleased to inform you that your manuscript has been judged scientifically suitable for publication and will be formally accepted for publication once it meets all outstanding technical requirements.

Kind regards,

Forough Mortazavi

Academic Editor

PLOS ONE
---

## [Editor Report · Acceptance letter]

29 Oct 2021

PONE-D-21-19421R2 

Internet addiction and sleep quality among medical students during the COVID-19 pandemic: A multinational cross-sectional survey 

Dear Dr. Pakpour:

I'm pleased to inform you that your manuscript has been deemed suitable for publication in PLOS ONE. Congratulations! Your manuscript is now with our production department. 

Kind regards, 

on behalf of

Dr. Forough Mortazavi 

Academic Editor

PLOS ONE